# CXCR2 Receptor: Regulation of Expression, Signal Transduction, and Involvement in Cancer

**DOI:** 10.3390/ijms23042168

**Published:** 2022-02-16

**Authors:** Jan Korbecki, Patrycja Kupnicka, Mikołaj Chlubek, Jarosław Gorący, Izabela Gutowska, Irena Baranowska-Bosiacka

**Affiliations:** 1Department of Biochemistry and Medical Chemistry, Pomeranian Medical University in Szczecin, Powstańców Wlkp. 72 Av., 70-111 Szczecin, Poland; jan.korbecki@onet.eu (J.K.); patrycja.kupnicka@pum.edu.pl (P.K.); mikolaj.chlubek@gmail.com (M.C.); 2Clinic of Cardiology, Pomeranian Medical University in Szczecin, Powstańców Wlkp. 72 Av., 70-111 Szczecin, Poland; jar.gor@pum.edu.pl; 3Department of Medical Chemistry, Pomeranian Medical University in Szczecin, Powstańców Wlkp. 72 Av., 70-111 Szczecin, Poland; izabela.gutowska@pum.edu.pl

**Keywords:** CXCR2, chemokine, interleukin-8, CXCL1, GRO-α, migration, MGSA, cancer

## Abstract

Chemokines are a group of about 50 chemotactic cytokines crucial for the migration of immune system cells and tumor cells, as well as for metastasis. One of the 20 chemokine receptors identified to date is CXCR2, a G-protein-coupled receptor (GPCR) whose most known ligands are CXCL8 (IL-8) and CXCL1 (GRO-α). In this article we present a comprehensive review of literature concerning the role of CXCR2 in cancer. We start with regulation of its expression at the transcriptional level and how this regulation involves microRNAs. We show the mechanism of CXCR2 signal transduction, in particular the action of heterotrimeric G proteins, phosphorylation, internalization, intracellular trafficking, sequestration, recycling, and degradation of CXCR2. We discuss in detail the mechanism of the effects of activated CXCR2 on the actin cytoskeleton. Finally, we describe the involvement of CXCR2 in cancer. We focused on the importance of CXCR2 in tumor processes such as proliferation, migration, and invasion of tumor cells as well as the effects of CXCR2 activation on angiogenesis, lymphangiogenesis, and cellular senescence. We also discuss the importance of CXCR2 in cell recruitment to the tumor niche including tumor-associated neutrophils (TAN), tumor-associated macrophages (TAM), myeloid-derived suppressor cells (MDSC), and regulatory T (T_reg_) cells.

## 1. Introduction

Forty-six human chemokines, grouped into four sub-families according to the number and spacing of two to four highly conserved N-terminal cysteines, are known to activate 20 chemokine receptors [1] and an additional 4 atypical chemokine receptors [2]. The names of the receptors correspond to the sub-family of chemokines they bind, with 10 receptors for CC chemokines, one receptor each for CX3C and XC chemokines, and 7 receptors for CXC chemokines [1]. One of the CXC receptors is CXC motif chemokine receptor 2 (CXCR2)—the receptor for the well-known studied chemokine CXC motif chemokine ligand 8 (CXCL8)/interleukin-8 (IL-8) [3].

These receptors are responsible for the properties of the chemokines. This is associated with the fact that a chemokine must activate its receptor in order to act. Signal transduction from such a receptor causes changes in the target cell, which are described as the action of a particular chemokine. Despite this, on PubMed (https://pubmed.ncbi.nlm.nih.gov; accessed on 30 December 2021), much more research is devoted to the chemokines than to their receptors, with more than 15 times as many papers on CXCL8/IL-8 than those about its receptors, CXC motif chemokine receptor 1 (CXCR1) and CXCR2. This is likely due to the fact that a given interaction between two cells is associated with the secretion of a ligand (chemokine) then an increase in the sensitivity of the target cell to the chemokines. For this reason, the importance of the chemokine receptors in biological processes is very often overlooked.

In an attempt to fill this gap, the aim of this review is to provide an insight into the activity of CXCR2—its expression, activation, signal transduction, and involvement in cancer.

## 2. *CXCR2* Gene and Regulation of Expression

### 2.1. CXCR2 Gene

CXCR2 was described for the first time in 1991 [4]. Its link to chemokines was discovered a year later when, together with CXCR1, it turned out to be a high affinity receptor for CXCL8/IL-8 [5]. CXCR2 was first named interleukin-8 receptor B (IL-8R-B); other variants of this name include IL-8R2 [6] and class II IL-8R [7].

The *CXCR2* and *CXCR1* genes are located at 2q34-q35 [8,9], forming a gene cluster with CXC motif chemokine receptor 2 pseudogene 1 (*CXCR2P1*). Proteins CXCR1 and CXCR2 show a 77% similarity in amino acid sequence [5,8], while the nucleotide sequence of *CXCR2P1* is 87% similar to *CXCR2* [9], although there are many frameshifts and stop codons in the sequence of *CXCR2P1* compared to *CXCR2*. All this information indicates that *CXCR1*, *CXCR2*, and *CXCR2P1* are derived from a common ancestral gene that has been duplicated.

### 2.2. Regulation of CXCR2 Expression

The promoter of the *CXCR2* gene contains a non-classical TATA-box which means that additional transcription initiation factors are required for the transcription of this gene [10]. Near the TATA-box are specificity protein 1 (Sp1) and activating protein-(AP-)2 binding sites [11]. Further away are CAAT-boxes, NF-IL6, AP-2, AP-3, and another Sp1 binding site [10]. The *CXCR2* gene promoter also contains two nuclear factor κB (NF-κB) binding sites at positions -331bp and -288bp [12].

*CXCR2* gene expression is elevated in hypoxia, being associated with the action of hypoxia inducible factor-1 (HIF-1) and NF-κB, as shown on prostate cancer cells [13]. However, the effect of hypoxia on *CXCR2* expression may depend on the selected model, such as in gastric cancer cells, where hypoxia reduces *CXCR2* expression [14].

*CXCR2* expression can be increased by signal transducer and activator of transcription 3 (STAT3), when attached to the *CXCR2* promoter [15,16,17]. Moreover, the *CXCR2* promoter can undergo histone acetylation on histone 3 at Lys^9^ (H3K9), which increases the expression of *CXCR2* in the spinal cord—a process crucial for the development of chronic pain after traumatic brain injury [18]. The promoter can also be attached to p53, increasing CXCR2 expression for cellular senescence.

The *CXCR2* gene consists of 11 exons [12]. The size of the open reading frame of this gene is 1065bp, and it is entirely located on the last exon [11]. The remaining exons form 11 different 5′-untranslated regions via alternative splicing [12]. So far, the significance of the formation of such a large number of different *CXCR2* mRNAs has not been understood—it may be related to the regulation of *CXCR2* expression. The 3′-UTR of CXCR2 mRNA also contains clustered adenine-uridine rich sequence elements (AREs), which indicate that the regulation of *CXCR2* expression may take place via the regulation of mRNA stability, although this has never been experimentally confirmed [11].

### 2.3. Involvement of MicroRNAs in the Regulation of CXCR2 Expression

Most studies on *CXCR2* promoter function were carried out in the 1990s and were not followed up with detailed research in the following decades. For this reason, information on this issue is somewhat limited.

Much better research was undertaken on the regulation of CXCR2 expression by miRNA, a very significant factor in cancer (Figure 1). However, we know only one miRNA involved in this process in the context of cancer, namely, miR-940 [19,20]. For example, a decrease in its expression in hepatocellular carcinoma was found to increase CXCR2 expression, which then leads to the increased migration of tumor cells [19]. A similar mechanism was demonstrated for tongue squamous cell carcinoma, where the downregulation of miR-940 expression increased CXCR2 expression and thus the response of these cancer cells to CXCR2 ligands.

The regulation of CXCR2 expression is important for maintaining endothelial integrity. Toxic agents such as oxidized low-density lipoprotein (oxLDL) increase the expression of metastasis associated lung adenocarcinoma transcript 1 (MALAT1), which sponges miR-22-3p, a miRNA that directly decreases CXCR2 expression [21].

Another miRNA, miR-155-5p, regulates CXCR2 expression during osteoclast differentiation [22]. Decreased expression of this miRNA results in an increase in CXCR2 expression, which leads to osteoclast differentiation, as shown in root resorption associated with orthodontic tooth movement.

In granulocytes and NK cells, CXCR2 expression is regulated by miR-4437 [23]. The expression of this miRNA is decreased in these cells in patients with ischemic stroke, and leads to increased expression of CXCR2 on granulocytes and NK cells, triggering the migration of these cells to the brain where they cause ischemic brain injury.

In silico analysis has shown some association of CXCR2 with atrial fibrillation—potential miRNAs that decrease CXCR2 expression are miR-5001-3p and miR-4283 [24]. Another in silico analysis of genes associated with neuropathic pain has shown that CXCR2 expression may be increased by reducing the expression of miR-7688-3p [25]. The study was associated with the earlier observation that CXCR2 is a receptor important in the development of neuropathic pain [26].

Some miRNAs that decrease CXCR2 expression may originate outside the human body. For example, preparations containing *Cordyceps militaris*, a fungus parasitizing on butterfly and moth caterpillars and that is used in traditional Chinese medicine, may decrease CXCR2 expression. The fungus contains miR-1321 and miR-3188—microRNAs that directly decrease CXCR2 expression [27].

It is also possible that miRNAs can influence CXCR2 activity. An example of this is miR-K12-3 [28], a miRNA from Kaposi’s sarcoma-associated herpesvirus (KSHV) that directly reduces the expression of G protein-coupled receptor kinase 2 (GRK2). The activity of this kinase leads to the desensitization and internalization of CXCR2 by phosphorylating this receptor [29]. The reduction in GRK2 expression increases the length of CXCR2 activation by ligands, a process crucial in Kaposi sarcoma development as it leads to angiogenesis and to KSHV latency.

## 3. CXCR2 Ligands

CXCR2 is a receptor for seven CXC chemokines: CXCL1, CXCL2, CXCL3, CXCL5, CXCL6, CXCL7, and CXCL8 [30,31]. Its strongest agonist is CXCL3 whose half maximal effective concentration (EC_50_) is 1 nM [30]; the weakest agonist is CXCL5 whose EC_50_ is 10.6 nM [30]. In contrast, chemokines CXCL1, CXCL2, CXCL6, CXCL7, and CXCL8 activate CXCR2 in the EC_50_ range 3 nM–7 nM [30,31]. CXCR2 activation by CXCL6 and CXCL8 occurs at the lowest concentrations required to activate CXCR1 [30,31]. On the other hand, some CXCR2 agonists are weak CXCR1 agonists, activating this receptor at concentrations above 40 nM. CXCR2 can also be activated by macrophage migration inhibitory factor (MIF) [32,33,34], although its affinity to CXCR2 is half that of CXCL8. The dissociation constant (K_d_) for MIF is 1.4 nM, while for CXCL8 it is 0.7 nM [32].

CXC chemokines that are ligands for CXCR2 can be divided into two groups: those that activate it at similar concentrations to CXCR1 activation (CXCL6 and CXCL8) and those at much higher concentrations (CXCL1, CXCL2, CXCL3, CXCL5, and CXCL7). Ligand binding by both these receptors occurs via the extracellularly located N-terminus, which determines ligand binding specificity [35]. The receptors show 77% similarity in terms of amino acid sequence [5,8] but the differences at the ends of the sequences result in differences in ligand binding specificity. Differences also occur in the binding of various CXC chemokines to CXCR2. The same or different amino acid residues may be responsible for the binding of individual CXC chemokines by CXCR1 and CXCR2. Asp^9^, Glu^12^, Lys^108^, and Lys^120^ are crucial for CXCL8 binding by CXCR2 [36], compared to Glu^7^, Asp^9^, and Glu^12^ for binding CXCL1. Differences in the binding of a particular chemokine are also related to differences between the chemokines. For example, significant differences between CXCL8 and CXCL1 in the loop preceding the first β-strand lead to a significantly different affinity of these two chemokines for the CXCR1 receptor [37]. At the same time, the residues responsible for binding a given chemokine may be different from the residues important in CXCR2 activation [36]. This indicates that the binding of the chemokine to CXCR2 is not associated with the activation of this receptor although both processes always occur at the same time.

Another poorly researched aspect of the CXCR2 function is the dimerization of the chemokine receptors. CXCR2 forms homodimers even in the absence of a ligand [38,39]. Homodimerization of CXCR2 occurs via the Ala^106^-Lys^163^ region [38]. It is also possible to heterodimerize CXCR1 with CXCR2 if these receptors are expressed simultaneously in the cell [39]. Ligands cause the breakdown of CXCR1:CXCR2 heterodimers but stabilize CXCR1:CXCR1 and CXCR2:CXCR2 homodimers, which then can undergo internalization [40]. Nevertheless, CXCR2 dimerization has been poorly researched and little is known about the role of this process in signal transduction from this receptor.

## 4. CXCR2 Signal Transduction

### 4.1. Heterotrimeric G Proteins and CXCR2

The CXCR2 receptor is a seven transmembrane G protein-coupled receptor (GPCR) due to the role heterotrimeric G proteins play in signal transduction from this receptor (Figure 2) [7,41]. CXCR2 couples to inhibitory guanine nucleotide regulatory protein (Gα_i_) [42,43,44,45,46]. As Gα_i_ is inhibited by pertussis toxin [47], signal transduction from CXCR2 is sensitive to pertussis toxin, yet it does not completely inhibit signal transduction. In particular, it does not completely inhibit Ca^2+^ flux which indicates the involvement of other G proteins or signaling pathways [47]. After CXCR2 activation and signal transduction on Gα_i_, heterotrimeric G protein dissociates into a heterodimer of Gβγ complex and Gα_i_ subunit [48]. Gβγ can cause PI3K activation and signal transduction via the phosphatidylinositol-4,5-bisphosphate 3-kinase (PI3K)-protein kinase B (PKB)/Akt pathway [49]. Gβγ can also activate phospholipase C-β (PLC-β) [50,51]. This enzyme generates diacylglycerol (DAG) and inositol triphosphate (IP_3_); the latter of these second messengers is responsible for Ca^2+^ mobilization. In turn, the Gα_i_ subunit is responsible for the inhibition of adenylyl cyclase [47,52,53,54]. With that said, the effect of CXCR2 on adenylyl cyclase appears to be a minor pathway in CXCR2 action compared to PI3K and PLC-β activation.

G_αi_ activity—and thus signal transduction from CXCR2—can be regulated. Immediately after CXCR2 activation, a G protein-coupled receptor kinase 6 (GRK6) complex is formed with the activator of G protein signaling 3 (AGS3) and Gα_i2_ [45]. The process of formation of this complex is dependent on the activation of Gα_i2_. This is followed by inhibition of the activity of Gα_i_. With this process of complex formation, phosphorylation of AGS3 by protein kinase C (PKC) occurs. However, the effect of this phosphorylation is not known. There are also other mechanisms that decrease CXCR2 activity. Moreover, the regulator of G protein signaling 5 (RGS5) inhibits CXCR2 activity in neutrophils and inactivates Gα_i_ [55].

Physical coupling of CXCR2 to Gα_i_ isoforms and the exact signal transduction is dependent on the cell type. For example, in neutrophils CXCR2 couples to Gα_i2_ and Gα_i3_ [44,45,46]. However, in human embryonic kidney (HEK)-293 cells transfected with CXCR2, this receptor couples to Gα_i2_ but not Gα_i1_ or Gα_i3_ [42,43]. Coupling with a given G protein is dependent on the expression level of the given Gα_i_ in the cell [46]. Importantly, various Gα_i_ have different roles in signal transduction from CXCR2. In neutrophils, Gα_i2_ is responsible for Ca^2+^ flux and neutrophil arrest in blood vessels [44,46]. In contrast, Gα_i3_ in neutrophils is responsible for neutrophil transmigration, migration, and activation of the PI3K-PKB/Akt pathway [46].

### 4.2. Phosphorylation of CXCR2

During signal transduction, CXCR2 becomes phosphorylated at the C-terminus residues Ser^342^, Ser^346^, Ser^347^, and Ser^348^ (Figure 3) which leads to desensitization, internalization, and sequestration of CXCR2 [7,56,57]. However, phosphorylation of CXCR2 at Ser^346^ and Ser^348^ does not induce internalization of CXCR2 [57]. CXCR2 phosphorylation is triggered by serine-threonine protein kinase: G protein-coupled receptor kinases (GRKs), in particular GRK2 [28,29] and GRK6 [58]. After internalization, CXCR2 undergoes dephosphorylation by protein phosphatase 2A (PP2A) [59,60]. This is followed by resensitization, i.e., the receptor is sensitive to its ligands again. Then it is transported back to the cell membrane.

CXCR2 can also undergo PKC-dependent desensitization [61]. In particular, activation of PKCε by CXCR1 and CCR5 causes cross-desensitization of CXCR2 [62]. In this process, PKCε phosphorylates the cytoplasmic tail of CXCR2. CXCR1 is the receptor for CXCL8, while CC motif chemokine receptor 5 (CCR5) is the receptor for CC motif chemokine ligand (CCL)3, CCL4, and CCL5 [1]. There are other mechanisms that decrease the activity of CXCR2. Therefore, these chemokines activate the CXCR1 and CCR5 receptors. This causes activation of PKCε, which phosphorylates and desensitizes CXCR2. In this way these chemokines disrupt CXCR2.

### 4.3. CXCR2 Chemosynapse

Many proteins interact with CXCR2 on the cytoplasmic side (Figure 4). Significantly, various proteins can attach to activated and non-activated CXCR2. Following activation of the receptor, some of these proteins stop directly binding to CXCR2, some start interacting with CXCR2, and some are still bound to CXCR2 regardless of its activation [63]. All proteins bound to CXCR2 have been called CXCR2 “chemosynapse” [63,64].

One of the proteins that interacts with CXCR2 is Na^+^/H^+^ exchanger regulatory factor-1 (NHERF1), an adaptor/scaffold protein [65]. Adaptor/scaffold proteins help in signal transduction by binding signaling pathway elements. NHERF1 has PDZ motifs that bind CXCR2 and PLC-β [65,66,67]. This interaction is essential for the activation of PLC-β by CXCR2 and thus signal transduction by this bound protein.

Another protein associated with the C-terminus of CXCR2 is IQ motif containing GTPase activating protein 1 (IQGAP1) [63]. This protein is associated with non-activated as well as with activated CXCR2. CXCR2 activation results in binding of activated cell division control protein 42 homolog (Cdc42) to IQGAP1. This interaction is important in Cdc42 activity because IQGAP1 inhibits the GTPase activity of Cdc42 and thus increases the activity of Cdc42 [68]. Moreover, CXCR2 activation results in F-actin binding to IQGAP1. Binding of actin filaments (F-actin) to IQGAP1 and Cdc42 activity leads to the formation of cross-link actin filaments which results in chemotaxis [63].

CXCR2 activation is followed by the attachment of vasodilator-stimulated phosphoprotein (VASP) to this receptor [63,69]. This is caused by the phosphorylation of Ser^239^ VASP by PKCδ and Ser^157^ by protein kinase A (PKA). The activation of PKCδ by CXCR2 most likely occurs through the activation of PLC-β by Gβγ [46,50,51]. PLC-β induces the generation of DAG and IP_3_, the former being the activator of PKCδ [70]. At the same time, PKCδ does not have a Ca^2+^-binding domain and thus Ca^2+^ flux does not affect the activity of this enzyme [71]. In contrast, the mechanism of PKA activation by CXCR2 is unclear. CXCR2 activation induces a signal transduction on Gα_i_ [42,43,44,45,46]. This G protein causes inhibition of adenylyl cyclase, thus reducing the generation of PKA activator cyclic adenosine monophosphate (cAMP) [47,53,54]. Although this means that CXCR2 should not increase but rather decrease PKA activity, it has been postulated that CXCR2 does increase cAMP generation through Ca^2+^ flux and activation of calcium-dependent adenylyl cyclase [46]. Upon phosphorylation, VASP begins to bind F-actin [69], resulting in VASP binding to CXCR2 and F-actin binding to the CXCR2 chemosynapse. This process is important in changes in the actin cytoskeleton under the influence of CXCR2 activity, resulting in chemotaxis.

The C-terminus of CXCR2 contains a leucine-rich domain—an LLKIL motif spanning 325–329aa of CXCR2 [72]. This domain is essential for the chemotactic response and is important but not essential in CXCR2 internalization and PKB/Akt and extracellular signal-regulated kinase (ERK) mitogen-activated protein kinase (MAPK) activation [73]. Various proteins bind to the domain, in particular, LIM and SH3 protein 1 (LASP-1) which bind independently of CXCR2 activation [74]. LASP-1 reduces basal c-Src activation and also participates in the activation of this protein. In turn, Src activates paxillin [74] and the engulfment and cell motility (ELMO)-dedicator of cytokinesis 2 (Dock2)- Rac family small GTPase 2 (Rac2) pathway [75] which is important in chemotaxis. Moreover, LASP-1 is important in the activation of the p130^CAS^-Rac1 pathway [74,76]. Rac1 [77] and Rac2 [75] can also be activated by CXCR2 through PI3K, which leads to further signal transduction as Rac family small GTPase 1 (Rac1) can activate p38 MAPK [78] and PAK [77]. LASP-1 is crucial for the activation of Cdc42-p21-activated kinase 1 (PAK1) cascade and ERK MAPK by CXCR2 [74,79]. Significantly, PAK1 activation is independent of ERK MAPK activation. All pathways in which LASP-1 participates affect cytoskeletal organization and thus cell migration.

In addition, the activation of CXCR2 is associated with the binding of Hsc70-interacting protein (Hip) to the leucine-rich domain at the C-terminus [72,73,80], resulting in the formation of a CXCR2-Hip-heat shock protein family A (Hsc70) complex essential for the internalization of CXCR2 and chemotaxis. CXCR2 activation also results in the binding of adaptor protein 2 (AP2) to the LLKIL motif [72,73,81]. By binding to membrane phosphatidylinositol phospholipid, AP2 docks in the plasma membrane. With the involvement of α-tubulin acetyl transferase, AP2 causes microtubule acetylation which is essential for directional cell locomotion [81,82].

Other proteins also start to interact with activated CXCR2, such as secretory carrier membrane protein 2 (SCAMP2), Nipsnap homolog 1, and dynein heavy chain 5, all important in intracellular transport [63]. Moreover, proteasome 20S subunit α 2 (PSMA2) and 26S proteosome subunit 9 begin to bind to activated CXCR2 [63].

Proteins that only interact with non-activated CXCR2 include P21-Arc, gelsolin, and plastin, important for the function of the actin cytoskeleton [63]. It is likely that these three proteins are uncoupled from CXCR2 upon activation of this receptor, which leads to changes in the actin cytoskeleton. Other proteins binding only to non-activated CXCR2 are Rab7, annexin 1, kinesin light chain-2, valosin-containing protein (VCP), important in intracellular transport. Furthermore, binding to inactivated CXCR2 are 14-3-3γ, an adaptor/scaffold protein, and Hsp90, Hsp75, and chaperonin TCP1 [63].

In contrast, some proteins bind to both activated and non-activated CXCR2 [63]. These include actin related protein 2/3 complex subunit 2, important in actin cytoskeleton function. Another protein from this group is 14-3-3ζ which is an adaptor/scaffold protein. Nevertheless, the role of the proteins mentioned in this subsection in signal transduction from CXCR2 is yet to be fully understood.

### 4.4. Other Pathways Activated by CXCR2

CXCR2 also activates pathways whose mechanisms are much less understood than the mentioned signaling pathways. For example, CXCR2 activates ERK MAPK via a variety of pathways. Activation of ERK MAPK by CXCR2 is sensitive to pertussis toxin, showing that it depends on G proteins which activate mitogen-activated protein kinase kinase 1 (MEK1) [52,83]. ERK MAPK activation by CXCR2 may also depend on Raf-MEK-ERK MAPK activation by activated STAT3 [15]. ERK MAPK activation may also be induced by reactive oxidant species (ROS) [84]. The generation of ROS in signal transduction from CXCR2 is dependent on NADPH oxidase activation—a process whose initiation is associated with Rac, activated by CXCR2 via PI3K [77]. ROS participates in the activation of c-Raf-MEK and subsequently ERK MAPK, as well as other MAPK cascades [84]. This entire process is inhibited by β-arrestin, preventing oxidative bursts and thus protecting the cell from death. β-arrestin also causes the inhibition of activation of other MAPK kinase cascades.

Activated CXCR2 can also indirectly activate ERK MAPK via the transactivation of epidermal growth factor receptor (EGFR) [85,86], in a mechanism identified in ovarian cancer cells.

CXCR2 also increases integrin β1 expression, which enhances focal adhesion kinase (FAK) activation [87]. At the same time, integrin activation may be induced by Ras-related protein 1 (Rap1) activation by CXCR2 [88]. FAK activation may also be associated with the PI3K-PKB/Akt or Gα_i_ subunit pathway, although this mechanism is yet to be confirmed [89]. FAK activation is followed by an increase in the expression of MMP2 and MMP9 which leads to the migration of tumor cells [89]. In particular, this process is important in lymphatic metastasis due to the secretion of CXCR2 ligands (CXCL1) by lymphatic endothelial cells.

CXCR2 activation also causes the phosphorylation of janus tyrosine kinase 2 (JAK2) and STAT3 [17,90]. The interaction between STAT3 and CXCR2 is a feed-forward loop where the activation of STAT3 increases the expression of CXCR2, which then activates STAT3 [16,17]. STAT3 activated by CXCR2 is important in the activation of the Raf-MEK-ERK MAPK pathway [15]. However, the very mechanism of STAT3 activation by CXCR2 is still unclear as no proteins relevant for direct activation of JAK2 bind directly to CXCR2 [63]. It is possible that CXCR2 may increase JAK2 expression which then may induce STAT3 activation [91]. In contrast, in granulocyte and macrophage progenitor cells (GMPs), CXCR2 activation results in decreased expression of Sin3-associated 18 kDa polypeptide (SAP18), which leads to an increase in PI3K expression and thus increased activation of ERK MAPK and STAT3 [92]. However, this mechanism of STAT3 activation appears to occur only in these cells.

CXCR2 has also been found to activate ras homolog family member A (RhoA), a representative of the Rho family of monomeric GTPases [93]. The activation of RhoA is important in endothelial cell chemotaxis and thus in angiogenesis induced by CXCR2 ligands. The activation of this protein by CXCR2 remains poorly understood.

### 4.5. Internalization of CXCR2

The aforementioned LLKIL motif is also important in the internalization of CXCR2 because it binds Hip [72,80] and AP2 [81], proteins involved not only in migration but also in the internalization of CXCR2. The latter protein, AP2, binds clathrin and β-arrestin and thus participates in the internalization of CXCR2 (Figure 5) [81]. CXCR2 also binds β-arrestin independently of AP2 [81,94] but is dependent on the phosphorylation of CXCR2. The internalization of CXCR2 requires the ubiquitination of Lys^327^ CXCR2 [95], a residue located in the aforementioned leucine-rich domain, which suggests that the ubiquitination is important in the activity of AP2 and Hip, although this needs to be verified.

The first step of CXCR2 internalization involves the formation of a clathrin-coated pit, while the subsequent steps involve Rab5, adaptin 2, and dynamin [59,94,96]. Nevertheless, CXCR2 internalization may occur independently of clathrin [81]. After internalization, CXCR2 is no longer in the cell membrane and is found in endosomes in the cytoplasm. This state is called sequestration. Importantly, internalization of CXCR2 is faster than that of CXCR1 [97]. However, recycling of CXCR2 takes much longer than CXCR1.

### 4.6. CXCR2 Recycling or Degradation

Upon activation, CXCR2 is internalized and then undergoes intracellular trafficking in a process dependent on Rab proteins [98]. The internalization of CXCR2 and formation of early endosomes with this receptor are dependent on clathrin, Rab5, and dynamin [96,99]. In this location, i.e., in the cytoplasm on endosomes, CXCR2 is not activated because it is no longer in the cell membrane—this state is called receptor sequestration. Subsequently, CXCR2 undergoes sorting decisions. It can be transported to recycling endosomes in a Rab11a-dependent process [99], and then recycled to the cell membrane, where it may become activated again and be involved in signal transduction. However, prolonged action by CXCR2 agonists is associated with a translocation of this receptor to early endosomes, then to late endosomes, and finally to lysosomes, which results in the degradation of CXCR2. Rab7 is responsible for directing CXCR2 into the degradation pathway [99]. Ras homolog B (RhoB), a GTPase activated by CXCR2, is responsible for sorting decisions [100]. GDP-bound RhoB induces the recycling of CXCR2, while GTP-bound RhoB is important in directing CXCR2 to lysosomes. In the absence of RhoB expression, the translocation of CXCR2 from endosomes to the cell membrane occurs via alternative recycling pathways, a part of the trans-Golgi network dependent on Rab4 and mannose-6-phosphate receptor.

## 5. The Role of CXCR2 in Cancer

CXCR2 has significant pro-tumor functions, where increased CXCR2 expression is associated with poorer patient prognosis, as shown by studies on acute myeloid leukemia [101], invasive ductal breast cancer [102], colorectal cancer [103], esophageal cancer [104], gastric cancer [16,17,105,106], intrahepatic cholangiocellular carcinoma [107], laryngeal squamous cell carcinoma [108], lung adenocarcinoma [109], non-small cell lung cancer [91], ovarian cancer [110,111], pancreatic ductal adenocarcinoma [112]. In contrast, in studies on triple-negative breast cancer [113], colorectal cancer [114], clear cell renal cell carcinoma [115], gastric cancer [116] and pancreatic ductal adenocarcinoma [117] (Table 1), no such association is shown, which may be related to the initiation of an antitumor immune response resulting in tumor infiltration by tumor-infiltrating lymphocytes (TILs), in particular CD8^+^ cytotoxic lymphocytes [113]. Additionally, CXCR2 activation may be significant in the anticancer functions of neutrophils [118].

In tumors, particularly in melanoma, CXCR2 is activated in an autocrine manner [119]. CXCR2-induced NF-κB activation increases the expression of the CXCR2 ligands which activate CXCR2. Activation of NF-κB via CXCR2 happens through a process dependent on Ras, mitogen-activated protein kinase kinase kinase 1 (MEKK1), and p38 [120]. Moreover, CXCR2-induced NF-κB activation occurs via the PI3K-PKB/Akt pathway [121] and via transforming growth factor (TGF)-β-activating kinase 1 (TAK1) [122]. At the same time, activation of this pathway may not so much directly depend on CXCR2 but rather on the transactivation of EGFR [121]. This activation of NF-κB leads to the expression of pro-inflammatory genes, including ligands for CXCR2 [119]. Moreover, the activation of NF-κB by CXCR2 causes cancer cell migration [121,122].

CXCR2 increases tumor cell proliferation in cancer. First of all, this happens via EGFR as a result of metalloprotease activation that cleaves heparin-binding epidermal growth factor-like growth factor (HB-EGF) [86], an EGFR ligand. HB-EGF, probably released via the actions of enzymes a disintegrin and metalloproteinase 17 (ADAM17) [123,124] or cathepsin B [125], induces the transactivation of EGFR, which leads to increased proliferation of epithelial ovarian cancer cells [86]. Second, the activation of CXCR2 results in increased expression of early growth response-1 (EGR-1) in esophageal cancer [126], which increases proliferation by elevating the expression of cyclin-dependent kinase 4 (CDK4). Third, CXCR2 activates PKB/Akt, which affects the action of p53 [127] via the activation of murine double minute 2 (Mdm2), an enzyme causing the ubiquitination and subsequent degradation of p53 [128]. Decreased expression of p53 results in decreased expression of p21, a cell cycle inhibitor—in this way CXCR2 activation results in decreased p21 expression and thus increased cell proliferation.

CXCR2 appears to inhibit proliferation in normal cells, which is associated with senescence dependent on p53 [129] and p38 MAPK [130]. Senescence itself causes an increase in the expression of CXCR2 and ligands for this receptor and thus autocrine enhancement of senescence [129,130]. If there is a mutation in the *TP53* gene or a decrease in p53 expression, then the described mechanism will not cause growth arrest but will stimulate proliferation [129], which is significant in cancer cell proliferation where p53 dysfunction is common [131]. The association of CXCR2 with senescence is important in the tumor microenvironment, as CXCR2 ligands secreted by cancer cells induce senescence of cancer-associated fibroblast (CAF) [132]. Then, these cells exhibit a senescence-associated secretory phenotype (SASP) that enhances cancer tumor growth [133]. In the absence of CXCR2 activation on CAFs, these cells become myofibroblasts [134].

CXCR2 activation also causes tumor cell migration. For example, it causes epithelial-mesenchymal transition (EMT). As a consequence of signal transduction on PI3K, the activation and accumulation of snail family transcriptional repressor 1 (Snail1) occur in the nucleus, which leads to the expression of EMT-related genes [135]. EMT may also involve the activation of STAT3 by CXCR2 [17]. CXCR2 is also significant in metastasis—if circulating tumor cells express CXCR2, then such cells will metastasize in organs with the high expression of ligands for this receptor. This activation of CXCR2 on a cancer cell in the blood is associated with an increase in vascular cell adhesion molecule-1 (VCAM-1) expression on cells with activated CXCR2, something that has been observed in the metastasis to the lung by osteosarcoma [136]. Moreover, CXCR2 may participate in bone metastasis, in particular in bone destruction during bone metastasis [137], something which is related to the recruitment and increased activity of osteoclasts. This is associated with the expression of CXCR2 on osteoclast precursors which migrate to a cancer cell releasing CXCR2 ligands [138,139]. Finally, one of the significant symptoms of bone cancer and bone metastases is bone pain, which is related to increased expression of CXCR2 ligands in the spinal cord [140,141,142]—CXCR2 activation in the spinal cord leads to bone cancer pain.

CXCR2 is involved in angiogenesis. For example, it is expressed on epithelial cells [143,144,145], which causes proliferation, tube formation, and migration of epithelial cells, leading to the formation of new blood vessels [143,145,146]. CXCR2 expression also occurs on lymphatic endothelial cells [147], which causes migration of these cells and tube formation, leading to the formation of new lymphatic vessels or lymphangiogenesis [147]. CXCR2 and its ligands are also part of the cascade of angiogenic factors. Vascular endothelial growth factor (VEGF), the most significant angiogenic factor, increases the expression of CXCL1 and CXCL8 in epithelial cells [148] and cancer cells [149]. VEGF and the aforementioned ELR^+^ CXC chemokines jointly participate in angiogenesis. CXCR2 activation also results in the secretion of epidermal growth factor (EGF) by epithelial cells [145], VEGF via STAT3 activation [90], and lymphangiogenic factors such as VEGF-C and VEGF-D by cancer cells [150]. The activation of EGFR on cancer cells also results in the activation of NF-κB and thus expression of pro-angiogenic CXCR2 ligands [151]. Additionally, CXCR2 is important in the mobilization and function of bone marrow-derived endothelial progenitor cells (EPCs) recruited to growing blood vessels [152].

CXCR2 is also significant in the recruitment of different cells to the tumor niche. CXCR2 ligands, upon binding to the receptor, induce recruitment and enhance the pro-tumor properties of tumor-associated macrophages (TAM) [153,154,155]. Recruitment of these cells is associated with CXCR2 expression on CD14 monocytes [156]. CXCR2 is also important in tumor-associated neutrophils (TAN) recruitment [153,157], with pro-tumorigenic N2 neutrophils showing increased CXCR2 expression relative to N1 polarized neutrophils [158]. Other cells that are recruited to the tumor niche via CXCR2 include granulocytic myeloid-derived suppressor cells (MDSCs) [153,159,160,161], regulatory T cells (T_reg_) [162], and bone marrow-derived mesenchymal cells (BM-MCs) [163]. CXCR2 ligands also indirectly induce the recruitment of monocytic MDSCs to the tumor niche. Increased levels of CXCR2 ligands result in the expansion of monocytic MDSCs in the bone marrow which increases the number of these cells [164].

CXCR2 plays an important role in cancer processes, a fact which can potentially be used in designing effective anti-cancer therapies. One possible course is the use of specific CXCR2 inhibitors to reduce the activity of this receptor [165,166,167,168]. This not only decreases cancer tumor growth but also increases the efficacy of the chemotherapy. In particular, due to the pro-angiogenic properties of CXCR2, inhibitors of this receptor increase the effectiveness of antiangiogenic therapy [169]. Another therapeutic approach is immunotherapy; as antiangiogenic lymphocytes lack CXCR2 receptor expression [170], they are not recruited to malignant tumors with a high concentration of CXCR2 ligands. CXCR2 expression may be increased via transduction in lymphocytes isolated from the patient [171,172,173] and when such transduced lymphocytes are reintroduced into the patient’s body, the cells will migrate to the tumor and destroy the tumor cells.

## 6. Perspective for Further Research

Current knowledge of CXCR2 is sufficient to understand the principles of the function and the importance of this receptor. However, some aspects require more comprehensive research. In particular, we know very little about the mechanisms regulating CXCR2 expression. Only at the outset of research on CXCR2 was the mechanism of regulation of the expression of this receptor described at the transcriptional level. Furthermore, very little is known about the mechanism of regulation of CXCR2 expression changes in mRNA stability—the CXCR2 mRNA sequence indicates that theoretically such regulation is possible although it is yet to be confirmed experimentally. There is also a considerable gap in knowledge on the significance of some proteins that directly interact with CXCR2 (14-3-3γ, 14-3-3ζ, Hsp90, Hsp75, and chaperonin TCP1). The biggest deficiency in current research on CXCR2 is the lack of research on the receptor itself and the focus on only one or two ligands despite the fact it is activated by seven different chemokines. Many research papers are content with only showing an increase in expression of one of the CXCR2 ligands, not paying any attention to the significance of CXCR2 itself and changes in its expression.

## Figures and Tables

**Figure 1 ijms-23-02168-f001:**
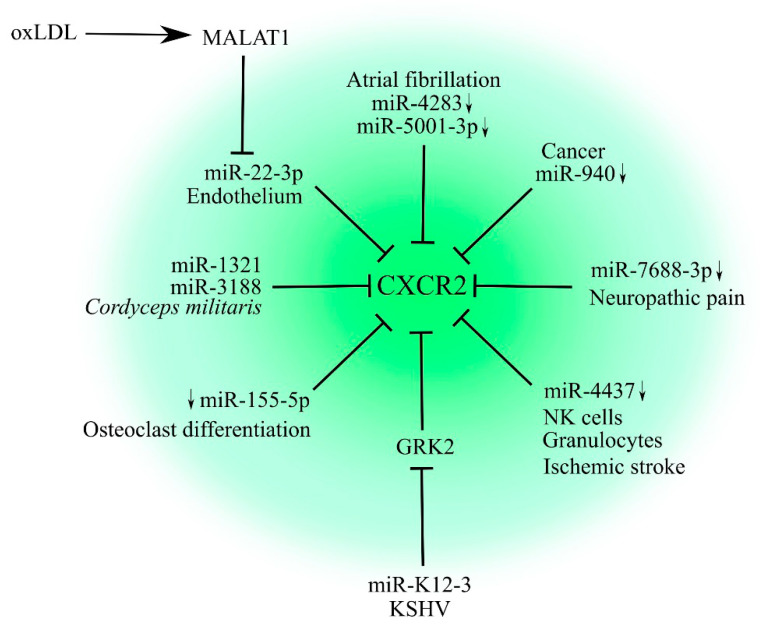
Regulation of CXCR2 by miRNAs. CXCR2 expression is regulated by miRNAs. Diseases such as cancer, ischemic stroke, atrial fibrillation, and neuropathic pain are associated with a decrease in the level of some miRNAs which results in a pathological increase in CXCR2 expression. Fungus *Cordyceps militaris* contains miRNAs that decrease the expression of CXCR2. The increased expression of CXCR2 can be physiological—it is often related to a decrease in the levels of certain miRNAs, resulting in an increase in CXCR2 expression and thus normal cell and tissue function. This is exemplified by osteoclast precursors, where increased CXCR2 expression leads to the differentiation of these cells into osteoclasts. This mechanism has been observed in root resorption associated with orthodontic tooth movement. Furthermore, increased CXCR2 expression on endothelial cells increases their proliferation which is associated with the preservation of endothelial integrity. CXCR2 activity is the effect of miRNAs on GRK2 expression. This mechanism is significant in cell infection by KSHV.

**Figure 2 ijms-23-02168-f002:**
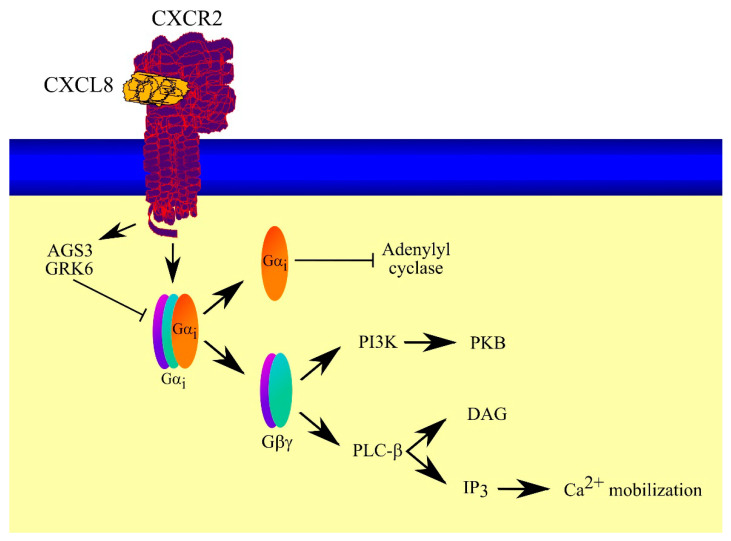
Consequences of G-protein activation by CXCR2. CXCR2 belongs to the seven transmembrane G-protein-coupled receptor family. CXCR2 is coupled with Gα_i_—upon CXCR2 activation, this G protein dissociates into heterodimer of Gβγ and Gα_i_ subunits. Gα_i_ subunit inhibits adenylyl cyclase activity, while Gβγ activates PI3K and PLC-β. Activation of PLC-β results in the production of DAG and IP_3_. IP_3_ induces the mobilization of Ca^2+^.

**Figure 3 ijms-23-02168-f003:**
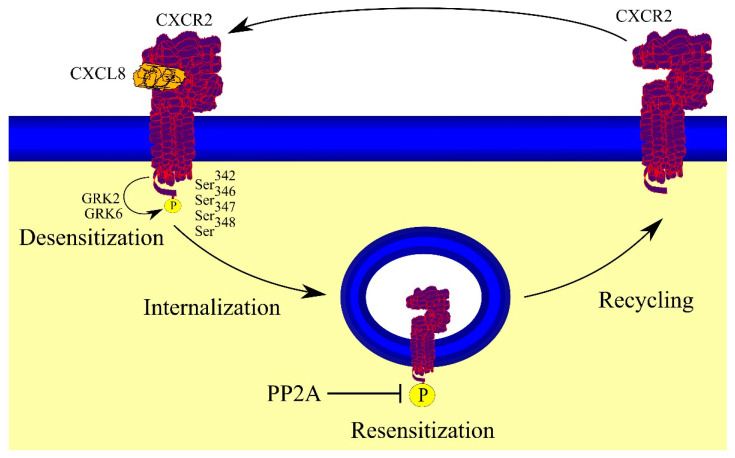
Importance of phosphorylation in the function of CXCR2 receptor. As a consequence of activation by its ligand, CXCR2 is phosphorylated at the C-terminus by GRK2 and GRK6. This leads to desensitization and subsequent internalization of CXCR2. In endosomes CXCR2 is dephosphorylated by PP2A, which leads to its resensitization and being recycled back to the cell membrane.

**Figure 4 ijms-23-02168-f004:**
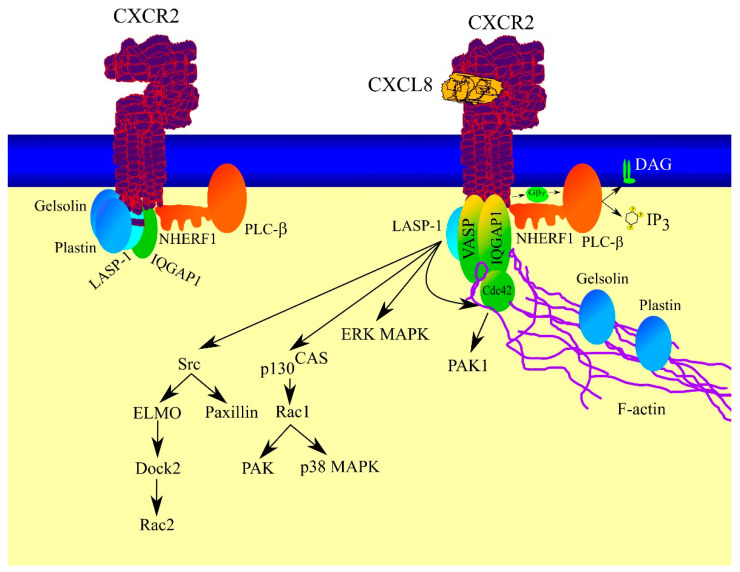
CXCR2 as a chemosynapse. Activated and inactivated CXCR2 bind various proteins which play an important role in signal transduction from this receptor. One such is NHERF1, which localizes PLC-β close to CXCR2—this allows the receptor to efficiently activate PLC-β. The activation of CXCR1 is followed by the dissociation of gelsolin and plastin, proteins which regulate the formation of F-actin close to the activated CXCR2 receptor. Moreover, upon CXCR2 activation, F-actin begins to bind to CXCR2 via VASP and IQGAP1—these processes lead to cell migration as a consequence of CXCR2 activation. Furthermore, CXCR2 activation mediated by LASP-1 is followed by the activation of various proteins including Cdc42, ERK MAPK, p130^CAS^, and Src.

**Figure 5 ijms-23-02168-f005:**
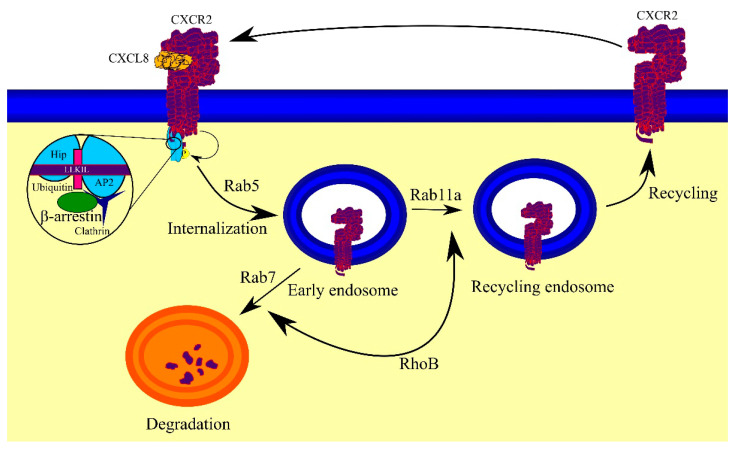
CXCR2 trafficking. Upon activation, CXCR2 is phosphorylated, and Hip and AP2 attach to the LLKIL motif at the C-terminus, which is also ubiquitinated. Clathrin and β-arrestin bind to AP2, resulting in the formation of a clathrin-coated pit. As a result, CXCR2 undergoes Rab5-dependent internalization into early endosomes. Subsequently, CXCR2 undergoes RhoB-dependent sorting decisions. CXCR2 can transition to recycling endosomes depending on Rab11a and then is transported back into the cell membrane. CXCR2 can also move to late endosomes and eventually to lysosomes in a Rab7-dependent process, which leads to the degradation of CXCR2.

**Table 1 ijms-23-02168-t001:** Effect of high CXCR2 expression in tumor on prognosis for patients with different cancers.

Type of Cancer	Impact on Prognosis	Number of Patients Selected for Testing	Notes	Source
Breast cancer: invasive ductal breast cancer	Worse prognosis	225	disease-free survival and overall survival	[102]
Breast cancer: triple-negative breast cancer	Better prognosis	290	recurrence-free survival and overall survival	[113]
Colorectal cancer	Not associated with patient prognosis	254	recurrence-free survival and overall survival	[114]
Colorectal cancer	Worse prognosis	134	overall survival and disease-free survival	[103]
Esophageal cancer	Worse prognosis	95	overall survival	[104]
Gastric cancer	Better prognosis	593	from TCGA database, overall survival	[116]
Gastric cancer	Worse prognosis	115	cumulative survival	[16]
Gastric cancer	Worse prognosis	269	overall survival	[106]
Gastric cancer	Worse prognosis	357	overall survival	[105]
Gastric cancer	Worse prognosis	155	overall survival	[17]
Intrahepatic cholangiocellular carcinoma	Worse prognosis	34	overall survival	[107]
Kidney cancer: clear cell renal cell carcinoma	Better prognosis	530	from TCGA database, overall survival	[115]
Laryngeal squamous cell carcinoma	Worse prognosis	109	overall survival	[108]
Leukemia: acute myeloid leukemia	Worse prognosis	45	recurrence-free survival and overall survival	[101]
Lung cancer: Lung adenocarcinoma	Worse prognosis	173	recurrence-free survival and overall survival	[109]
Lung cancer: non-small cell lung cancer	Worse prognosis	340	disease-free survival and overall survival	[91]
Ovarian cancer	Worse prognosis	240	disease-free survival	[110]
Ovarian cancer	Worse prognosis	370	from TCGA database, overall survival	[111]
Pancreatic ductal adenocarcinoma	Not associated with patient prognosis	102	recurrence-free survival and overall survival	[117]
Pancreatic ductal adenocarcinoma	Worse prognosis	44	overall survival	[112]

Red background—increased CXCR2 expression in the tumor is associated with a worse prognosis; blue background—increased CXCR2 expression in the tumor is associated with a better prognosis; gray background—CXCR2 expression in the tumor is not associated with patient prognosis.

## Data Availability

Not applicable.

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
