# Peer review of "CXCR2 Receptor: Regulation of Expression, Signal Transduction, and Involvement in Cancer"

_ijms, 2022, doi:10.3390/ijms23042168_

Round 1

Reviewer 1 Report

General comments to the paper entitled

CXCR2 receptor: regulation of expression, signal transduction and involvement in cancer. A literature review

The presented paper is an excellent review work. Focus on CXCR2 receptor and introduce all the related molecular processes in the receptor possesses an important role. The content of the paper is well structured.

I suggest a couple of minor changes

line: 237

Draw an arrow on Fig 3. from right to left to close the cycle on the upper part of the figure

Suggest the same in Fig. 5.

line: 435

Increased should change for decreased

There is some contradiction in Table 1. because the arrow is indicating the impact on prognosis but also the increase or decrease of CXCR2 expression.

Author Response

Rev1

line: 237

Draw an arrow on Fig 3. from right to left to close the cycle on the upper part of the figure

Suggest the same in Fig. 5.

The arrows have been added to the two figures according to recommendation.

line: 435

Increased should change for decreased

We are very grateful for this remark. We changed the markings in the table because the arrows introduced big problems with interpretation. The table describes the effect of increased CXCR2 expression on prognosis. The red background shows that increased CXCR2 expression worsens the prognosis. A blue background is increased expression of CXCR2 improves the prognosis.

There is some contradiction in Table 1. because the arrow is indicating the impact on prognosis but also the increase or decrease of CXCR2 expression.

We are very grateful for this remark. The arrows can be problematic as they indicate the level of expression in most manuscripts. Therefore, they were replaced with the background colour and description.

Reviewer 2 Report

Congratulations to the authors for writing this manuscript. The work appears well constructed and detailed. In fact, the manuscript provides notions of basic biology, also carefully describing the studies carried out with CXCR2 in cancer and laying the foundations for future research.

Moreover, considering that there are not many publications in this field, it represents an innovative work and therefore worthy of being published in IJMS.

I only suggest few changes/corrections.

MINOR POINTS

- Delete “A literature review” from the title.

- According to the IJMS Authors Guidelines the abstract should not exceed 200 words, please check. I think a few words need to be deleted to meet the journal's standards

- Line 37-39, the sentence need a supporting reference

- Line 358-361, the sentence need a supporting reference

- The table is well done, however I would suggest a reduction of the characters in the titles of the first line, so as not to break the words. (Example: prog-nosis etc)

- Line 438-440, this part appears a bit confused, I recommend a remodeling of the sentences

Author Response

Rev2

- Delete “A literature review” from the title.

The title was corrected as recommended

- According to the IJMS Authors Guidelines the abstract should not exceed 200 words, please check. I think a few words need to be deleted to meet the journal's standards

The abstract was corrected in line with the recommendations. The abstract now consists of 199 words.

- Line 37-39, the sentence need a supporting reference

The manuscript was corrected as recommended, references have been added

- Line 358-361, the sentence need a supporting reference

The manuscript was corrected as recommended, references have been added

- The table is well done, however I would suggest a reduction of the characters in the titles of the first line, so as not to break the words. (Example: prog-nosis etc)

The tables and whole manuscript was corrected as recommended

 - Line 438-440, this part appears a bit confused, I recommend a remodeling of the sentences

The sentences were divided into shorter ones as recommended
